# Iron and Iodine Status in Pregnant Women from A Developing Country and Its Relation to Pregnancy Outcomes

**DOI:** 10.3390/ijerph16224414

**Published:** 2019-11-11

**Authors:** Sehar Iqbal, Petra Rust, Lisbeth Weitensfelder, Inayat Ali, Michael Kundi, Hanns Moshammer, Cem Ekmekcioglu

**Affiliations:** 1Department of Environmental Health, Center for Public Health, Medical University of Vienna, Kinderspitalgasse 15, 1090 Vienna, Austria; saheriqbal55@gmail.com (S.I.); lisbeth.weitensfelder@meduniwien.ac.at (L.W.); michael.kundi@meduniwien.ac.at (M.K.); cem.ekmekcioglu@meduniwien.ac.at (C.E.); 2Department of Nutritional Sciences, University of Vienna, Althanstrasse 14, 1090 Vienna, Austria; petra.rust@univie.ac.at; 3Department of Social and Cultural Anthropology, University of Vienna, Universitätsstrasse 7, 1010 Vienna, Austria; inayat_qau@yahoo.com

**Keywords:** iron, iodine, Hb, thyroid hormones, pregnancy complications

## Abstract

Birth related complications and comorbidities are highly associated with a poor nutritional status of pregnant women, whereas iron and iodine are among especially important trace elements for healthy maternal and fetal outcomes. The study compares the status of iron, iodine, and related functional parameters in pregnant and non-pregnant women from a developing country and associates the data with pregnancy complications. The concentrations of ferritin, hemoglobin (Hb), total triiodothyronine (TT3), total thyroxine (TT4), and thyroid-stimulating hormone (TSH) were determined in the blood serum of 80 pregnant women at the time of delivery and compared with 40 non-pregnant healthy controls. Spot urine samples were taken to evaluate the urinary iodine concentration (UIC). In pregnant women, ferritin, Hb concentrations, and UIC were significantly lower, and TT4 values were significantly higher compared to controls. Higher Hb levels were tendentially associated with a reduced risk for pregnancy complications (OR = 0.747, CI (95%) 0.556–1.004; *p* = 0.053). Regarding covariates, only previous miscarriages were marginally associated with pregnancy complications. High consumption of dairy products was associated with lower Hb and ferritin values. Our results suggest that pregnant women from a developing country have lower iron status with Hb levels being possibly associated with pregnancy complications.

## 1. Introduction

Birth related complications and comorbidities are highly associated with poor nutritional status of pregnant women [1] with a balanced dietary intake and adequate nutrition playing a vital role in normal bodily functions and healthy fetal growth [2]. Trace elements, although required in a minute quantity, are important to activate multiple transporters, transcription factors, and enzymes in the body [3]. These elements are essential for different biochemical processes and to prevent tissue and cellular damage [4]. 

Iron is well-known for its functions in oxygen transport, in the immune system, and in sustaining normal neuronal function [5]. Particularly during pregnancy, the need for iron increases due to the expansion of plasma volume, red blood cell (RBC) mass, and erythropoietic activity, as well as the fetal growth and accumulation of iron stores for the first six month of life [6]. Iron is vital for the normal growth of the fetus, for healthy maternal-fetal outcomes, and to prevent maternal and fetal iron deficiency anemia [1]. Therefore, iron-rich diets and supplements are recommended during pregnancy for the prevention and treatment of maternal anemia and adverse fetal outcomes [7,8]. However, iron supplementation may be problematic in iron-replete women, when increased hemoglobin or ferritin levels appear to be associated with an increased risk of gestational diabetes mellitus [7], since, iron overload has an important role in pathogenesis of diabetes through an increase in oxidative stress, β-cell failure, and insulin resistance [9].

In addition to iron, iodine is also an important micronutrient, which is needed for the synthesis of thyroid hormones and, therefore, essential for normal fetal neurodevelopment [10]. An adequate iodine intake is a pre-requisite for healthy gestation due to increases in maternal-fetus thyroid hormone production and an increase in maternal renal iodine losses [11]. However, in Eastern European and African countries, 55% of pregnant women have deficient urine iodine levels (UIC < 50 µg/L) [12]. 

The data in the present study derive from Pakistan, which is a developing country facing important public health problems such as malnutrition and micronutrient deficiencies. A national nutrition survey of Pakistan revealed that 25.9% of the total pregnant women and 19.9% of non-pregnant women had iron deficiency anemia, and 47.7% were suffering from iodine deficiency [13]. According to the Pakistan Demographic and Health Survey 2006–2007, 1 among 89 women died due to pregnancy complications in the country, and as much as 25.6 percent of births were reported with low birth weights (less than 2.5 kg) [14]. The high-level mortality rate and neonatal deaths are the results of an ongoing interplay among various factors with the National Nutrition Survey 2018 indicating widespread micronutrient deficiencies among the most vulnerable population groups of children under 5 years and women of reproductive age [15,16]. 

The scarcity of data regarding the iron and iodine status among pregnant and non-pregnant women from Pakistan and its relationship with pregnancy outcomes necessitates filling this gap. Therefore, we compared the iron and iodine status and the related functional variables in pregnant women and non-pregnant women and also related the data to pregnancy outcomes and food intake.

## 2. Materials and Methods 

### 2.1. Study Population

For the study, 80 healthy pregnant women (26 ± 4 years of age) at the time of delivery were recruited from the District Hospital Khanewal, Pakistan. Another 40, age-matched, healthy, non-pregnant women 25 ± 4 years old from the same district were invited to participate in the study. Characteristics of study participants are shown in Table 1. An informed consent form was signed by each participant, and the study was approved by the National Bioethics Committee Pakistan (ref no. 4-87/NBC-281/17/1439). 

### 2.2. Confounding Factors and Food Frequency Questionnaire

Variables such as age, socio-demographic, and lifestyle factors may influence the risk for micronutrient deficiencies in mother and child. The participants were asked to provide information about their educational and socioeconomic status, lifestyle, smoking status (all participants were non-smokers), physical activity, dietary habits (no participant was on a diet), previous miscarriages, pregnancy complications, and antenatal visits. Physical activity was evaluated by using the American College of Obstetricians and Gynecologists (ACOG) physical activity guidelines 2015 (target of ≥20 min of exercise on most days/week) [17]. BMI was measured at the time of delivery, and women taking iron or iodine supplements were excluded from the study. However, five pregnant women reportedly were taking folic acid supplements during pregnancy. 

In addition, to obtain data about dietary habits, a 17-item food frequency questionnaire (FFQ) was administered by personal interview of each participant. The FFQ included questions about the consumption of meat, fish, processed food, dairy products, eggs, grains, rice or noodles, legumes and pulses, vegetables, fruits, salad, oils, nuts, sweets, snacks, and cold and hot drinks.

### 2.3. Pregnancy Outcomes

Two instructed nurses from hospital staff of the gynecology department were responsible for recording pregnancy outcomes such as prenatal mortality, duration of gestation, miscarriages, premature rupture of membranes, intrauterine growth restriction, preterm birth, and placental weight and in addition infant anthropometrics, which included birth weight, head circumference, appearance (skin color), pulse (heart rate), grimace response (reflexes), activity (muscle tone), and respiration. From the last 5 parameters, the APGAR (**A**ppearance, **P**ulse, **G**rimace, **A**ctivity, **R**espiration) score was calculated. Each criterion scales from zero to two, resulting in scores ranging from zero to 10. Scores of 7 and above are considered as normal; 4 to 6, fairly low; and 3 and below as critically low.

### 2.4. Laboratory Analyses

Qualified nurses from the district hospital Khanewal had collected the blood samples of pregnant women at the time of delivery and 24 h before delivery in the case of a recommended cesarean section. Forty age-matched healthy women, who visited the hospital for routine obstetrical checkups and had negative pregnancy tests were asked to participate in the study as a control group, and well-trained instructed nurses from the hospital staff collected their blood samples for analyses. 

Serum ferritin and hemoglobin (Hb) were used to measure iron and anemia status of the participants, respectively. Blood samples were collected in EDTA vials for the analysis of Hb with an automated hematology analyzer (Sysmex XP-300; Sysmex America Inc., Lincolnshire, IL, USA). Blood samples for the analysis of ferritin and thyroid profile were taken in separate gel vials and centrifuged at 3000 rpm for 10 min to separate serum. Serum samples were stored in freeze cups at −20 °C until analyses. Serum ferritin was measured by enzyme-linked fluorescent immunoassay (ELFA) performed in an automated instrument (VIDAS^®^ FERRITIN ref. 30411, bioMérieux Inc., Durham, NC, USA). 

Iodine status was assessed by analyzing iodine in urine samples, and measuring total triiodothyronine (TT3), total thyroxine (TT4), and thyroid-stimulating hormone (TSH) in the blood serum. Spot urine samples were collected in screw-capped plastic containers and stored at −20 °C until the analysis was completed. Samples were transported in cold packs to the UVAS laboratory. Urinary iodine concentration (UIC) was analyzed in duplicate by using the ammonium persulfate digestion method, and the thyroid profile was analyzed by enzyme-linked immunosorbent assay (ELISA) kits provided by CALBIOTECH, Germany (catalog no PT-T3-96, PT-T4-96, and TS227T).

### 2.5. Statistical Methods

Since no prior data were available about micronutrient status, we performed sample size calculations based on the prerequisite to detect an effect size of Cohen’s *d* = 0.7 at the overall significance level of 5% with a power of 80% and a ratio of 2:1 for pregnant women to controls. The required sample sizes were determined as 77 and 39 and rounded to 80 and 40. This sample size for pregnant women was sufficient to detect an odds ratio of 2.5 for the combined pregnancy and child complications.

All micronutrient data underwent a preliminary analysis of variance, including the primary independent variables together with the covariates. The residuals of these analyses were tested for normality applying Kolmogorov–Smirnov tests with Lilliefors corrected *p*-values. Logarithmic transformation was necessary for TT3 and TT4 because of significant deviations from normality. Iron and iodine concentrations were assessed by general linear model analyses controlled for education, income, and physical activity for comparison of pregnant women and non-pregnant women and for comparisons between primi- and multipara, additionally controlled for age, antenatal visits, and previous miscarriages. As multiple pregnancies represent a state of increased nutritional requirements resulting in a greater maternal nutrient drain and an accelerated depletion of nutritional reserves, we also compared primi- with multipara. 

Binary logistic regression analyses were performed to analyze the association of the iron and iodine concentration and anemia with pregnancy complications. For this, all pregnancy complications (maternal + fetal/neonatal) were fused and included as the dependent variable (yes/no). Trace element concentrations were used as the independent variable and age, income status, education levels, physical activity, antenatal visits, and previous miscarriages were taken as covariates. Data were presented as odds ratios with 95% confidence intervals, and a *p*-value of <0.05 was considered to represent statistical significance.

A stepwise multiple linear regression analysis was performed to assess the impact of the intake of different foods on the iron and iodine status in pregnant women. Logistic stepwise regression was performed with anemia as the dependent variable. For this purpose, data from the FFQ were standardized to weekly intake data. Significant coefficients were transformed into percent increase/decrease of the dependent variable by an increase of the intake by one unit (one additional uptake per week).

All statistical analyses were performed by SPSS, version 25 (IBM Corp., Armonk, NY, USA).

## 3. Results 

The socio-demographic data showed that, amongst others, 56.3% of pregnant women and 47.5% control women had no education. Further, 17.5% of pregnant women reported low physical activity, and most of the pregnant women were housewives as compared to controls (Table 1). In addition, the pregnancy spacing in multipara was 1.88 ± 1.01 years (mean ± SD).

Based on the cut-off values for Hb from WHO (<10 g/dL), 68.8% of pregnant women were found to be anemic, with 6.3% having severe, 37.5% moderate, and 25.0% mild anemia. Regarding ferritin and iodine status, 56.3% showed depleted iron storage and 68.8% had insufficient UIC, respectively.

Ferritin and Hb levels were significantly lower in pregnant women (15.09 ± 5.59 µg/L and 10.12 ± 1.85 g/dL, respectively) as compared to the non-pregnant group (24.14 ± 3.46 µg/L and 11.71 ± 1.11 g/dL, respectively, both *p* < 0.001, Table 2). 

For iodine related parameters, levels of TSH and TT3 did not statistically differ between groups (Table 2). However, TT4 (0.80 ± 0.08 µg/dL in pregnant vs 0.74 ± 0.03 µg/dL in non-pregnant women) and urinary iodine levels (114.06 ± 35.99 µg/L in pregnant vs 127.83 ± 40.01 µg/L in non-pregnant women) were significantly different in pregnant women (Table 2).

In a sub-group analysis of pregnant women, we compared primipara with multipara and found no difference in the iron and iodine status between these two groups (Table 3).

### 3.1. Association of Iron or Iodine Status during Pregnancy with the Risk of Pregnancy Complications

A total of 26 (32.5%) participants had one or more pregnancy complications (maternal and child) such as premature rupture of membranes, bleeding, hypertension, small for gestational age, stillbirth, preterm birth, low birth weight, and low APGAR scores, which were registered during data collection (Table 4). 

Binary logistic regression analyses were used to determine the association of iron and iodine status with all pregnancy complications. Higher Hb levels were associated with lower odds ratios, though not reaching statistical significance, while for other variables, no association was found (Table 5). From the covariates, only the history of previous miscarriage(s) was marginally associated with an increased OR for pregnancy complications (OR = 2.827 (1.000; 8.013); *p* = 0.050). 

### 3.2. Food Intake and Correlation of Different Foods with Iron and Iodine status

In Figure 1, the average intake per week of different foods and beverages in both groups is shown. The study groups especially consumed grains, oils, and also dairy products, while the weekly intake of meat, fish, but also sweets and snacks such as halwa, kheer, and samosa were relatively low compared to grains (wheat), fruits, eggs, and dairy products (Figure 1).

Compared to the pregnant women, 30% of women of the control group ate meat 1–3 times a week and 15% ate meat 4–7 times a week, while, in the group of pregnant women, only 10% consumed meat 1–3 times a week, and 7.5% ate 4–7 times a week. Furthermore, consumption of dairy products was higher in the group of pregnant women (67.5% once/day, and 25.1% more than 2 times a day) compared to controls (55% once/day). Fish consumption, as an essential source of iodine, was low in both study groups, and one in two pregnant women never ate fish. Only 2.5% of women in each study group reached the recommendation of 2 portions per week (Figure 1).

To evaluate the intake of different foods during pregnancy in relation to the concentration of iron, iodine, and the related variables, a stepwise multiple linear regression analysis was performed. While most of the food items were not correlated with iron and iodine status in the pregnant women, increased consumption of dairy products was associated with an approximately 20% increased risk of anemia (per one portion per week). Moreover, increased consumption of dairy products was associated with decreased Hb and ferritin (only significant results shown in Table 6). Increased consumption of processed food was associated with increased levels of ferritin and decreased levels of TT3. 

## 4. Discussion

Overall, a high percentage of anemia (68.8%) and urinary iodine deficiency (68.8%) were found in the pregnant women of our study, which is in line with previous studies showing the widespread iron and iodine deficiency in the country [18,19]. In our study, we found lower concentrations of Hb and serum ferritin in pregnant women compared to non-pregnant women. This finding is in agreement with previous findings, which showed, due to physiological changes, a gradual decrease of Hb and ferritin concentration with the progression of pregnancy [20,21]. Regarding thyroid hormones, we found higher concentrations of TT4 (TSH and TT3 n.s. higher) in pregnant women compared to the non-pregnant group. This is also in line with previous studies, which recorded a significant increase in the mean TT4 concentration during the third trimester compared with the mean of non-pregnant women and showed an overall increase in the 2.5th, 50th, and 97.5th percentile of TT3 and TT4 during pregnancy as compared to non-pregnant women [22,23]. 

Measurement of urinary iodine is an established method to assess the iodine status of populations [24]. Urinary iodine levels were significantly lower in pregnant women in our study. Similarly, a study in Sri Lanka showed a gradual decrease in median UIC with the progression of pregnancy [25], while an investigation from Iran reported no significant differences between pregnant and non-pregnant women [26]. Gestational changes in thyroid physiology are widely influenced by the iodine status of pregnant women due to an increase in renal iodine excretion, thyroxine-binding proteins, and an increase in thyroid hormone production [27]. Therefore, the WHO recommends the median UIC of 150–249 µg/L for adequate iodine intake during pregnancy [24]. 

Regarding maternal and fetal/neonatal pregnancy complications, we observed that higher Hb levels were tendentially associated with a reduced risk for pregnancy complications. Good iron supply and normal Hb values have been shown to be protective against perinatal and neonatal mortality and to improve maternal hematologic status as well [28]. The other iron and iodine variables were not associated with pregnancy complications in our study. However, it is known that a compromised thyroid status during pregnancy may be related to adverse pregnancy outcomes [29], and UICs below 50 µg/L in the third trimester increase the risk for small for gestational age infants [30]. Similarly, the prevalence of anemia in pregnant women during the last trimester appears to be associated with a decrease in mean birth weight as compared to controls [31]. In addition to pregnancy itself, further major risk factors for (iron deficiency) anemia are a diet low in bioavailable iron, like vegan diets, and gastrointestinal bleedings [32,33]. No participant suffered from gastrointestinal bleedings. However, there was some indication that the pregnant women of our study might have lower (bioavailable) dietary iron intake than the non-pregnant women, with lower meat intake and higher intakes of dairy products and grains, which could have affected iron absorption at least to some extent. 

We found that the previous miscarriage history was associated with higher odds for pregnancy complications. This is in agreement with previous studies showing that women with histories of recurrent miscarriages have a higher risk of low APGAR scores, small for gestational age, preterm delivery, and antepartum hemorrhage [34]. It is estimated that 70% of conceptions are lost prior to a live birth as a consequence of different etiologies, including parental chromosomal anomalies, maternal thrombophilic disorders, endocrine disturbances, and immune dysfunctions [35]. Furthermore, micronutrient deficiencies, subclinical hypothyroidism, and thyroid autoimmunity have been found to be associated with an increased risk of miscarriage [1,36]. However, a nationwide follow-up study in Denmark concluded that 25.2% of the miscarriages were preventable by modification of multiple risk factors to low-risk levels, such as maternal weight and age at conception as potentially modifiable pre-pregnant risk factors, and alcohol consumption, lifting of >20 kg daily, and night work during pregnancy. Modification of all risk factors acting before pregnancy could prevent 14.7%, while risk factors during pregnancy helped to prevent about 12.5% of miscarriages [37]. 

By applying an FFQ, we found that women from Pakistan especially ate (wheat) grains, which is the basic food consumed in the country. Not merely meat, nuts but also fish consumption in our sample was low, with the latter possibly being problematic due to the importance of long-chain omega-3 fatty acids, like docosahexaenoic acid (DHA), for good neurological development of the fetus. With few exceptions, we did not find significant correlations between different foods and the iron or iodine variables. A significant association was found for the intake of vegetables and eggs with Hb and TT3 (Table 6). A recent study [38] also found that a higher intake of green leafy vegetables and eggs is positively associated with Hb concentration. However, another investigation did not detect a correlation of vegetable intake with an increase in Hb levels during pregnancy [39]. Our study did not find any correlation between meat and fish intake with iron and iodine status, respectively, although, for example, a positive association between meat intake and iron status was well-documented in women of reproductive age [40]. However, it has to be noted that these nutritional components were almost lacking in our study groups (see Figure 1). A previous study in Pakistan found that dietary practices such as consumption of red meat more than twice a week tended to have an anemia protective effect, while tea consumption and non-food items (pica) have been associated with an increased risk for anemia during pregnancy [18]. In developing countries, animal foods are expensive and scarce, resulting in limited access and intake. Of note is the significant relationship between dairy products’ consumption and both indicators of iron status, Hb and ferritin. According to former studies, it is suggested that calcium from dairy products can reduce the absorption of (non-hem) iron in the gut (reviewed in [41]), although the results are conflicting [42], and the effect of calcium supplementation on iron absorption is dose-dependent [43]. We also found that increased processed food consumption was significantly related to increased ferritin concentrations and decreased TT3 concentrations, and increased snack food intake was related to increased TSH. Nevertheless, we cannot explain these associations due to limited literature in the field.

### Study Limitations

Our study has some limitations. It is known that some sociodemographic and lifestyle variables can influence the indicators of iron and iodine status [44]. Although we controlled for major covariates, we cannot exclude that certain confounders have played at least some role in the outcomes of the study. Another limitation might be that thyroid function is different per trimester and also in comparison with pre-pregnancy. However, to obtain consistent results between the pregnant women, we measured thyroid function shortly before delivery at the end of the third trimester, in accordance with other studies [45,46]. Furthermore, since Pakistani pregnant women have poor antenatal care-seeking behavior, it would have been very difficult to assess iron and thyroid function throughout pregnancy. We did not assess urinary creatinine concentration, and therefore, it was not possible to estimate the hydration status of the participants. This can be counted as a certain limitation. However, also in other studies, this was not accomplished [25,26]. Since the sample size for each maternal or child complication was relatively low, we pooled all complications into one variable. This could be regarded as another potential limitation. Altogether, five pregnant women were reported taking folic acid supplements during pregnancy, while information regarding multivitamins before pregnancy was not recorded and considered as a certain limitation. However, it is known that women in Pakistan seldomly use multivitamin supplements [13].

## 5. Conclusions

In summary, our study showed lower ferritin, Hb, and UIC concentrations in pregnant women as compared to non-pregnant controls, whereas reported dietary habits do not sufficiently support the elevated needs in pregnancy.

Gaps in feto–maternal health and nutritional health coverage remain a great public health problem in developing countries unless appropriate interventions are designed. Therefore, the WHO recommended daily oral iron and folic acid supplementation with 30 mg to 60 mg of elemental iron and 400 µg (0.4 mg) of folic acid for pregnant women to prevent maternal anemia, puerperal sepsis, low birth weight, and preterm birth [47]. Additionally, in areas of moderate and severe iodine deficiency where median urinary iodine is less than 50 µg/L or total goiter rate is more than 20%, additional iodine in the form of a supplement should be given to all pregnant and lactating women [48]. Furthermore, a daily iodine intake of 250 µg is recommended from WHO for pregnant and lactating women [49]. 

A pregnant woman should start pregnancy with an iron reserve of at least 500 mg to meet the increased iron demands of pregnancy, yet only 20% of reproductive-aged women are estimated to have this iron store [50]. Similarly, approximately 2 billion people are affected by iodine deficiency in the world [51]. Hence, comprehensive planning to strengthen the iodized salt programs, maternal iron supplementation, guarantee of adequate maternal and child health services, reduced food insecurity, targeting rural areas (to reach remote areas as well), community engagement, and health education may be some important predictors to combat iron and iodine deficiency in developing countries including Pakistan. Furthermore, pre-screening of the thyroid profile before the start of pregnancy would also help to minimize the iodine related birth disorders. 

This is probably the first Pakistani study that emphasized the association of iron and iodine status with pregnancy and perinatal outcomes. Further studies are needed to clarify the role of food intake on the status of iron and iodine and the reasons for the sub-optimal micronutrient status in pregnant women in Pakistan. 

## Figures and Tables

**Figure 1 ijerph-16-04414-f001:**
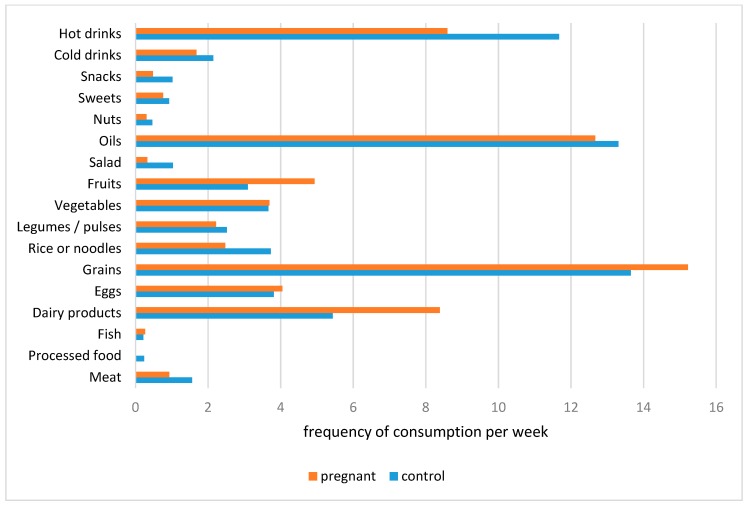
Weekly frequency of consumption of different types of foods and beverages (pregnant vs. control group).

**Table 1 ijerph-16-04414-t001:** Characteristics of study participants (mean ± SD or *n* (%)).

		Controls	Pregnant Women	Primigravid	Multigravid
Characteristic	Category/Unit	*n* = 40	*n* = 80	*n* = 40	*n* = 40
Age	Years	25 ± 4	26 ± 4	24 ± 4	28 ± 3
Education	No education	19 (47.5%)	45 (56.3%)	19 (47.5%)	26 (65.0%)
	Primary education	6 (15.0%)	30 (37.5%)	17 (42.5%)	13 (32.5%)
	High school or more	15 (37.5%)	5 (6.3%)	4 (10.0%)	1 (2.5%)
Occupation	Housewife	22 (55.0%)	78 (97.5%)	38 (95.0%)	40 (100.0%)
	Working	18 (45.0%)	2 (2.5%)	2 (5.0%)	0 (0.0%)
Income	$/month	599 ± 692	183 ± 157	181 ± 161	186 ± 154
Marital status	Married	22 (55.0%)	80 (100.0%)	40 (100.0%)	40 (100.0%)
Family type	Single family	25 (62.5%)	21 (26.3%)	9 (22.5%)	12 (30.0%)
	Joint family	15 (37.5%)	59 (73.8%)	31 (77.5%)	28 (70.0%)
Children	Number	1.7 ± 2.0	1.6 ± 2.1	-	3.3 ± 1.8
Previous miscarriages	Yes	0 (0.0%)	27 (33.8%)	9 (22.5%)	18 (45.0%)
Physical activity	Low	18 (45.0%)	14 (17.5%)	8 (20.0%)	6 (15.0%)
	Moderate/high	22 (55.0%)	66 (82.5%)	32 (80.0%)	34 (85.0%)
BMI ^a^	kg/m^2^	25.6 ± 4.0	27.0 ± 2.9	25.9 ± 3.0	28.1 ± 2.5

^a^ BMI (body mass index) was recorded at the time of delivery.

**Table 2 ijerph-16-04414-t002:** Concentrations of Hb in whole blood, ferritin, TSH, TT3, TT4 in blood serum and iodine in spot urine in pregnant women as compared to the control group.

Variable	Control Group Mean ± SD Md (IQR)	Pregnant Group Mean ± SD Md (IQR)	95% Confidence Interval of Adjusted Mean Difference	*p*-Value ^a^
Hb (g/dL)	11.71 ± 1.11 11.90 (11.20–12.40]	10.12 ± 1.85 10.10 (9.08–11.23]	(−2.43; −1.06)	<0.001
Ferritin (µg/L)	24.14 ± 3.46 24.00 (21.50–26.63)	15.09 ± 5.59 14.50 (11.38–18.00)	(−11.14; −7.01)	<0.001
TSH (µIU/mL)	5.05 ± 7.71 2.97 (1.75–4.41)	5.22 ± 4.81 4.06 (2.60–5.57)	(−2.15; 2.61)	0.970
lg-TT3 ^b^ (ng/mL)	0.33 ± 0.21 0.32 (0.24–0.48)	0.40 ± 0.27 0.41 (0.23–0.64)	(−0.05; 0.16)	0.257
lg-TT4 ^b^ (µg/dL)	0.74 ± 0.03 0.73 (0.73–0.74)	0.80 ± 0.08 0.79 (0.73–0.86)	(0.03; 0.08)	<0.001
Urinary iodine ^c^ (µg/L)	127.83 ± 40.01 118.50 (100.75–143.25)	114.06 ± 35.99 108.00 (91.00–130.00)	(−35.66; −0.12)	0.013
Anemia ᵈ (%)	7.5%	68.8%		<0.001

SD standard deviation, Md median, IQR interquartile range; ^a^ General linear model controlled for education, income, and physical activity; ^b^ TT3 and TT4 were logarithmically transformed for analysis; ^c^ Cut-off UIC < 150 µg/L; ᵈ Cut-off Hb < 10 g/dL and/or ferritin < 13 µg/L.

**Table 3 ijerph-16-04414-t003:** Concentrations of Hb in whole blood, ferritin, TSH, TT3, TT4 in blood serum and iodine in spot urine in pregnant women (primipara vs. multipara).

Variable	Primipara Mean ± SD Md (IQR)	MultiparaMean ± SD Md (IQR)	95% Confidence Interval of Adjusted Mean Difference	*p*-Value ^a^
Hb (g/dL)	10.04 ± 1.87 10.05 (9.08–11.20)	10.21 ± 1.86 10.30 (9.25–11.40)	(−0.19; 1.65)	0.118
Ferritin (µg/L)	15.57 ± 5.89 14.50 (12.50–17.13)	14.62 ± 5.29 14.25 (10.50–18.13)	(−4.90; 0.69)	0.157
TSH (µIU/mL)	4.88 ± 4.75 3.80 (2.74–5.13)	5.56 ± 4.89 4.17 (2.60–7.35)	(−2.13; 2.60)	0.873
lg-TT3 ^b^ (ng/mL)	0.40 ± 0.28 0.44 (0.19–0.64)	0.39 ± 0.27 0.37 (0.26–0.59)	(−0.11; 0.16)	0.626
lg-TT4 ^b^ (µg/dL)	0.80 ± 0.08 0.79 (0.73–0.85)	0.80 ± 0.08 0.78 (0.73–0.86)	(−0.06; 0.01)	0.102
Urinary iodine (µg/L)	116.59 ± 39.68 111.00 (82.75–134.75)	111.61 ± 32.45 106.00 (98.00–118.00)	(−21.24; 18.80)	0.979
Anemia ^c^ (%)	67.5%	70.0%		0.500

SD standard deviation, Md median, IQR interquartile range; ^a^ General linear model controlled for age, education, income, physical activity, antenatal visits and previous miscarriages; ^b^ TT3 and TT4 were logarithmically transformed for analysis; ^c^ Cut-off Hb < 10 g/dL.

**Table 4 ijerph-16-04414-t004:** Registered number (%) of pregnancy complications (maternal and child).

Pregnancy Complications ^a^	26 (32.5%)
Type of Complication	*n* (% of Whole Pregnant Sample) of Complications
Premature rupture of membranes	4 (5%)
Bleeding	7 (8.8%)
Hypertension,	5 (6.3%)
Small for gestational age	7 (8.8%)
Stillbirth	4 (5%)
Preterm birth	4 (5%)
Low birth weight	10 (12.6%)
Low APGAR score	7 (8.8%)

^a^ recorded 1 or more pregnancy complication (maternal and child).

**Table 5 ijerph-16-04414-t005:** Association of Hb in whole blood, ferritin, TSH, T3, T4 in blood serum and iodine in spot urine with the risk of pregnancy complications (any maternal and child complication) ^a,b^.

Variable.	Odds Ratio	95% Confidence Interval	*p*-Value
Hb	0.747	(0.556, 1.004)	0.053
Ferritin	1.020	(0.936, 1.111)	0.648
TSH	0.985	(0.881, 1.101)	0.786
lg-TT3 ^c^	0.548	(0.083, 3.600)	0.531
lg-TT4 ^c^	0.020	(0.000, 29.644)	0.293
Urine iodine	1.010	(0.994, 1.027)	0.230
Anemia ^d^	0.691	(0.242, 1.977)	0.491

^a^ dependent variable was any pregnancy complication (yes/no); ^b^ controlled for age, education, income, antenatal visits, previous miscarriage, and physical activity; ^c^ TT3 and TT4 were logarithmically transformed due to deviation from normality; ^d^ Hb < 10 g/dL.

**Table 6 ijerph-16-04414-t006:** Stepwise multiple linear regression analyzing the association of the intake of different foods and beverages on the status of iron and iodine ^a,b^.

Variable	β-Coefficient	% Increase/Decrease by One Portion/Week	*p*-Value
Anemia ^c^			
Dairy products	0.166	21.3	0.041
Fruits	0.272	30.2	0.025
Hb			
Vegetables	0.260	1.0	<0.001
Eggs	0.129	0.5	0.031
Dairy products	−0.144	−0.6	<0.001
Oils	−0.113	−0.4	0.009
Ferritin			
Dairy products	−0.012	−2.8	0.001
Processed food	0.109	28.5	0.013
Rice and noodles	0.015	3.5	0.025
lg-TT3			
Vegetables	0.028	6.7	0.019
Eggs	0.025	5.9	0.009
Processed food	−0.180	−51.4	0.009
lg-TT4			
Oils	0.005	1.1	0.013
Eggs	−0.006	−1.4	0.024
TSH			
Snacks (sweet + salty)	1.493	29.0	0.005
Meat	−0.737	−14.3	0.035
Urinary iodine			
Sweets	9.087	7.6	0.014

^a^ only significant results are presented; ^b^ controlled for age, BMI, income, physical activity, and marital status; ^c^ Hb < 10 g/dL.

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
