# Peer review of "Iron and Iodine Status in Pregnant Women from A Developing Country and Its Relation to Pregnancy Outcomes"

_ijerph, 2019, doi:10.3390/ijerph16224414_

Round 1

Reviewer 1 Report

Abstract: The authors should state the cut-off p-value they used for their analysis in the statistical section. If a two sided test <0.05 was considered significant, then the association of higher Hb level with lower risk of pregnancy complication is not significant and only suggestive by the p-value and CI.

Table 1. Please add to the footnote that BMI is at the time of delivery. 

Table 4 and P6 L174: "Maternal or Child" or "Maternal and Child"? Please be consistent.

Conclusion: please consider revising lower iron and iodine status. It's clear what lower status means.

Author Response

Comments and Suggestions for Authors

This research looks at associations between certain determinants of pregnancy outcome. Iodine and iron deficiencies are major players. I have the following comments, suggested edits and questions.

Reviewer’s comments:

Abstract: The authors should state the cut-off p-value they used for their analysis in the statistical section. If a two-sided test <0.05 was considered significant, then the association of higher Hb level with lower risk of pregnancy complication is not significant and only suggestive by the p-value and CI.

Ok. Thank you. We added a new sentence in the statistical section (line 141). “a p-value of <0.05 was considered to represent statistical significance”.

The sentence regarding association of higher Hb level with lower risk of pregnancy complication was changed as “Higher Hb levels were tendentially associated with a reduced risk for pregnancy complications (OR = 0.747, CI [95%] 0.556-1.004; p = 0.053).”

Table 1. Please add to the footnote that BMI is at the time of delivery.

Ok. Thank you. We added the information in the footnotes.

Table 4 and P6 L174: "Maternal or Child" or "Maternal and Child"? Please be consistent.

Ok. Thank you. We checked and edited.

Conclusion: please consider revising lower iron and iodine status. It's clear what lower status means.

Thank you for the valuable comment. As it is clear what lower iron and iodine status means, we revised the conclusions. I.e. In summary, our study showed lower ferritin, Hb and UIC concentrations in pregnant women as compared to non-pregnant controls, whereas reported dietary habits do not sufficiently support the elevated needs in pregnancy.

Reviewer 2 Report

This is just a descriptive study of iron and iodine status in pregnant and non-pregnant women from Pakistan. The associations made by the authors in the paper are not substantiated by the results. The sample size is too small to make any statement about the associations between the nutrient intake and birth outcomes. This paper would be better presented as describing intake and status in this vulnerable population. 

Introduction: The introduction is missing critical information on the controversy of iron supplementation in pregnancy, especially among iron replete women. This discussion should include the data which indicates a possible connection with the development of GDM.

Methods: Sample size is too small to make inferences about iron or iodine status with birth outcomes. For example, preterm birth rate is 5% (n=4). It is not clear what variable this study is powered on. The authors indicate a power analysis was performed, but which variable is not clearly defined.

The FFQ is a limited tool for dietary data assessment. A 24-hour diet recall in conjunction with the FFQ is a better choice and provides more data. 

There is no mention of pregnancy spacing in the manuscript. This is a  critical variable to iron status.

The manuscript is further limited by it's cross-sectional nature (at delivery). This does not provide data on iron and iodine status throughout pregnancy. This is especially important if associations to birth outcomes are being made. 

In summary, this manuscript needs to be reframed as a descriptive study at delivery of pregnant and non-pregnant women's iron and iodine status in Pakistan. Any associations made to birth outcomes are inappropriate. 

Author Response

Reviewer 2

Comments and Suggestions for Authors

This is just a descriptive study of iron and iodine status in pregnant and non-pregnant women from Pakistan. The associations made by the authors in the paper are not substantiated by the results. The sample size is too small to make any statement about the associations between the nutrient intake and birth outcomes. This paper would be better presented as describing intake and status in this vulnerable population.

Introduction: The introduction is missing critical information on the controversy of iron supplementation in pregnancy, especially among iron replete women. This discussion should include the data which indicates a possible connection with the development of GDM.

Thank you for the valuable input and suggestion. We updated/incorporated the introduction section accordingly.

Line 44. Therefore, iron rich diets and supplements are recommended during pregnancy for the prevention and treatment of maternal anemia and adverse fetal outcomes [7,8]. However, iron supplementation may be problematic in iron replete women, when increased hemoglobin or ferritin levels appear to be associated with an increased risk of gestational diabetes mellitus [7], since, iron overload has an important role in pathogenesis of diabetes through an increase in oxidative stress, β-cell failure and insulin resistance [9].

[Iqbal S, Ekmekcioglu C (2019) Maternal and neonatal outcomes related to iron supplementation or iron status: a summary of meta-analyses. J Matern Neonatal Med 32:1528–1540]. https://doi.org/10.1080/14767058.2017.1406915]. [Imdad A, Bhutta ZA (2012) Routine iron/folate supplementation during pregnancy: Effect on maternal anaemia and birth outcomes. Paediatr Perinat Epidemiol 26:168–177. https://doi.org/10.1111/j.1365-3016.2012.01312.x]. [Simcox JA, McClain DA (2013) Iron and Diabetes Risk. Cell Metab 17:329–341. https://doi.org/10.1037/a0032811.Child]

Methods: Sample size is too small to make inferences about iron or iodine status with birth outcomes. For example, preterm birth rate is 5% (n=4). It is not clear what variable this study is powered on. The authors indicate a power analysis was performed, but which variable is not clearly defined.

Thank you. We pooled all maternal and child complications to have enough power for statistical analyses. All maternal and child complications were fused into one variable (yes/no) and sample size was enough to detect an effect size of Cohen’s d=0.7 at the overall significance level of 5 % with a power of 80 %.

The FFQ is a limited tool for dietary data assessment. A 24-hour diet recall in conjunction with the FFQ is a better choice and provides more data.

Thanks for this valuable comment. A 24-hour dietary recall was unfortunately not assessed. FFQ data give information on frequency of consumption of different food groups and can be compared with recommendations of the food based dietary guidelines, which are based on nutrient based guidelines. Therefore, we calculated percentage of women in each group according to recommendations of the food based dietary guidelines for iron and iodine.

There is no mention of pregnancy spacing in the manuscript. This is a critical variable to iron status.

Ok agreed. We added this information in the results section (line 153). “The pregnancy spacing in multipara was 1.88 ± 1.01 years (mean±SD)”.

The manuscript is further limited by its cross-sectional nature (at delivery). This does not provide data on iron and iodine status throughout pregnancy. This is especially important if associations to birth outcomes are being made. 

Thank you. Yes, it is true that the status of iron and iodine was not assessed throughout pregnancy We added related literature and discussed the related topic in the discussion: “…and UICs below 50 µg/l at the third trimester increase the risk for small for gestational age infants [30]. Similarly, prevalence of anemia in pregnant women during the last trimester appear to be associated with a decrease mean birth weight as compared to controls [31]”.  [Demmouche A, Lazrag A, Moulessehoul S (2011) Prevalence of anaemia in pregnant women during the last trimester: Consequense for birth weight. Eur Rev Med Pharmacol Sci 15:436–445]

We additionally discussed this point as a potential limitation including the rationale for measuring trace element levels at delivery.

“Another limitation might be that thyroid function is different per trimester and also in comparison with pre-pregnancy. However, to obtain consistent results between the pregnant women, we measured thyroid function shortly before delivery at the end of the third trimester, in accordance with other studies [45, 46]. Furthermore, since Pakistani pregnant women have a poor antenatal care seeking behaviour, it would have been very difficult to assess iron and thyroid function throughout pregnancy”

Reviewer 3 Report

This is an interesting and well-written paper and I thank the authors for the opportunity to review it. The paper describes the roles of the micro-nutrients iron and iodine in health and addresses a lack of data in Pakistan with regards to iron and iodine status and related markers in pregnant and non-pregnant women. The authors also explore relationships of iron and iodine status/markers with maternal/infant health outcomes and food intake. Measurement of a number of iron and iodine related markers is a strength of this work. There are some minor grammar and typological errors in the manuscript. Some points for consideration:

1. Abstract line 24-25: Higher Hb is reported to be associated with lower occurrence of pregnancy complications. However, this is of borderline/non-significance which should be acknowledged in abstract. 

2. Line 47-49. I am not sure what is meant by ‘an excessive demand’ – does this refer to the high requirement/need for iodine during pregnancy?

3. Line 79 and Table 1. How was physical activity defined and assessed? Levels of physical activity appear high, particularly in pregnant women. Including some detail on physical activity measurement in the methods may be helpful.

4. Table 1. Pregnant and control women appear to be quite different in demographic characteristics such as education, occupation and income, which may challenge direct comparisons between the two groups. Is this a reflection on how the control group were recruited? The following should be considered: a) a statistical analysis of such differences, b) inclusion of some detail on participant demographics in the results and c) the discussion should include some reflection as to how demographic differences may influence study findings. Such differences may be particularly pertinent to food intake data?

5. Line 187 and abstract. History of previous miscarriage is reported to have been associated with an increase in pregnancy complications – what is the P-value?

6. Table 6. While a negative relationship between dairy products and iron status is known, and the lack of relationships between iron and meat and iodine and fish in this study are well presented and discussed, other relationships between iron and iodine status/markers and food group intakes which may not have a clear pathway are not discussed e.g. increased processed food consumption was significantly related to increased ferritin concentrations and decreased TT3 concentrations and increased snack food intake was related to increased TSH. Why might these relationships have been found? Might food intake data reflect overall dietary patterns in addition to intakes of specific food groups? Is there a public health relevance to such findings or might they be spurious?

7. Maternal and infant pregnancy complications/health outcomes were collated for analysis. While understandable in order to increase power in this relatively small sample, this approach may have implications/limitations given that each outcome may have a different aetiology and mechanistic pathway and an individual relationship (or none) with iron/iodine status and markers, as opposed to one overarching link between all maternal/infant outcomes and iron/iodine status and markers. This should be acknowledged in the manuscript.

Author Response

Reviewer 3:

Comments and Suggestions for Authors

This is an interesting and well-written paper and I thank the authors for the opportunity to review it. The paper describes the roles of the micro-nutrients iron and iodine in health and addresses a lack of data in Pakistan with regards to iron and iodine status and related markers in pregnant and non-pregnant women. The authors also explore relationships of iron and iodine status/markers with maternal/infant health outcomes and food intake. Measurement of a number of iron and iodine related markers is a strength of this work. There are some minor grammar and typological errors in the manuscript. Some points for consideration:

Reviewer’s comments:

Abstract line 24-25: Higher Hb is reported to be associated with lower occurrence of pregnancy complications. However, this is of borderline/non-significance which should be acknowledged in abstract. 

Ok. Thank you for the comment. We revised the sentence, “Higher Hb levels were tendentially associated with a reduced risk for pregnancy complications (OR = 0.747, CI [95%] 0.556-1.004; p = 0.053).”

Line 47-49. I am not sure what is meant by ‘an excessive demand’ – does this refer to the high requirement/need for iodine during pregnancy?

Ok. For more clarity, we revised the sentence (line 51-53). “An adequate iodine intake is a pre-requisite for healthy gestation due to increases in maternal-fetus thyroid hormone production and an increase in maternal renal iodine losses”.

Line 79 and Table 1. How was physical activity defined and assessed? Levels of physical activity appear high, particularly in pregnant women. Including some detail on physical activity measurement in the methods may be helpful.

Thank you for your valuable suggestion. We added the information in the methods section (line 84): “Physical activity was evaluated through using ACOG Physical Activity Guidelines 2015 [17].

Table 1. Pregnant and control women appear to be quite different in demographic characteristics such as education, occupation and income, which may challenge direct comparisons between the two groups. Is this a reflection on how the control group were recruited? The following should be considered: a) a statistical analysis of such differences, b) inclusion of some detail on participant demographics in the results and c) the discussion should include some reflection as to how demographic differences may influence study findings. Such differences may be particularly pertinent to food intake data?

Thanks for your valuable comments. We added additional information in the beginning of the results section. i.e.

“The socio-demographic data showed that, amongst others, 56.3 % of pregnant women and 47.5% control women had no education. 17.5% pregnant women reported of low physical activity, and most of the pregnant women were housewives as compared to controls (Table 1). In addition, the pregnancy spacing in multipara was 1.88 ± 1.01 years (mean±SD).

Further, we discussed the difference and influence of demographic characteristics as a potential limitation in the limitation section:

“It is known that that some sociodemographic and lifestyle variables can influence the indicators of iron and iodine status [44]. Although we controlled for major covariates we cannot exclude, that certain confounders have played at least some roles in the outcomes of the study.”

[ Pfeiffer CM, Sternberg MR, Caldwell KL, Pan Y. Race-Ethnicity Is Related to Biomarkers of Iron and Iodine Status after Adjusting for Sociodemographic and Lifestyle Variables in NHANES 2003–2006. J Nutr 2013; 143:977S-985S].

Line 187 and abstract. History of previous miscarriage is reported to have been associated with an increase in pregnancy complications – what is the P-value?

Ok.  we added the information (line 197). “(OR = 2.827 [1.000; 8.013], p=0.050). We added the term “marginally” to abstract and results.

Table 6. While a negative relationship between dairy products and iron status is known, and the lack of relationships between iron and meat and iodine and fish in this study are well presented and discussed, other relationships between iron and iodine status/markers and food group intakes which may not have a clear pathway are not discussed e.g. increased processed food consumption was significantly related to increased ferritin concentrations and decreased TT3 concentrations and increased snack food intake was related to increased TSH. Why might these relationships have been found? Might food intake data reflect overall dietary patterns in addition to intakes of specific food groups? Is there a public health relevance to such findings or might they be spurious?

Thanks for the suggestion. However, the reasons for these associations are rather unknown and speculative. Also, the public health relevance is difficult to translate. You are right, food intake data do not necessarily reflect overall dietary patterns but they are a valuable tool to obtain general food intake habits which are, amongst others, important to analyse and optimise healthy nutrition patterns.

We added information in the discussion before the limitation section.

“We also found that increased processed food consumption was significantly related to increased ferritin concentrations and decreased TT3 concentrations and increased snack food intake was related to increased TSH. Nevertheless, we cannot explain these association due to limited literature in the field.”.

Maternal and infant pregnancy complications/health outcomes were collated for analysis. While understandable in order to increase power in this relatively small sample, this approach may have implications/limitations given that each outcome may have a different aetiology and mechanistic pathway and an individual relationship (or none) with iron/iodine status and markers, as opposed to one overarching link between all maternal/infant outcomes and iron/iodine status and markers. This should be acknowledged in the manuscript.

Ok. Agreed. Therefore, we added this information in the limitation section.

“Since the sample size for each maternal and child complication was relatively low, we pooled all complications into one variable. This could be regarded as another potential limitation”.

This manuscript is a resubmission of an earlier submission. The following is a list of the peer review reports and author responses from that submission.

Round 1

Reviewer 1 Report

 Iron and iodine status in pregnant women from a developing country and its relation to pregnancy outcomes

This research looks at associations between certain determinants of pregnancy outcome. Iodine and iron deficiencies are major players. I have the following comments, suggested edits and questions.

Reviewer’s comments:

Abstract, lines 24-25: “Higher Hb levels were non-significantly associated with a reduced risk for pregnancy complications”. Please replace “non-significantly” with “not significantly”. Edit as such through the article.

Abstract, lines 24-25: “Higher Hb levels were non-significantly associated with a reduced risk for pregnancy complications” is conceptually problematic. Having a normal level does not pose a risk of complications. The statement will be better made if stated in relation to Hb deficiency level or anemia. Such as “Having low Hb level was associated (or not associated) with pregnancy complications”.

Abstract, lines 29-30: “Consumption of dairy products may impair bioavailability of iron”. The authors cannot conclude or assert this because the study did not test that. At best, the authors can assert that dairy products are low in iron.

Methods, line 98: Hemoglobin (Hb) concentration is not specific for iron status, unless microcytic anemia is present. Ferritin is a good measure of iron status. You could say anemia status for Hb.

Methods, lines 105-111: It was not clear if hydration status was corrected. One way to do this is to include urinary creatinine concentration in the regression models to adjust for hydration status. If hydration status was not corrected, state it as a limitation as part of the discussion section.

Methods, lines 129-130: Which trace elements were studied? If it is just iron an iodine, just name them in place of trace elements. Edit all in this article.

Methods, lines 129-134: The authors should add texts to show how the trace elements (Hb and iodine and ferritin?) were treated in the logistic regression analyses. Were they grouped into low/high, and if so, what health cut-offs were used?

Results: Expecting to see what proportions (%) were anemic, iron deficient per ferritin, and iodine deficient. Without using these health cut-offs to check associations, it will be difficult to interpret the results in terms of significance to health. Analyses of the associations between anemia, iron deficiency, and iodine deficiency with the observed pregnancy outcomes are expected and must be included. I thought the odds ratios (likelihood ratios) were for those, but apparently not.

Also add as a footnote, the health cut-offs (Hb cut-off for anemia in pregnancy, cut-off for ferritin, and iodine) under the relevant tables – Table 1 or Table 2, as applicable.

Results, Table 5: “B-value” should be replaced with “β-coefficient”.

Results: Some of the frequently consumed sweets and snacks should be mentioned in the appropriate section of the results.

I guess the discussion section will be updated in relation to the suggested edits.

Add strengths and limitations as a new paragraph just before the conclusions section so you can add limitations due to some deficiencies in this study including those suggestec above.

Reviewer 2 Report

Iqbal and colleagues in a cross-sectional study of 80 and 40 pregnant and non-pregnant women, respectively, investigated the status iron and iodine and related functional assessment at delivery concerning adverse maternal and fetal pregnancy outcomes in Pakistan. There are major considerations regarding the methods and the results.

1. P1L24-26: What does “non-statistically associated with” mean? If it is not statistically significant at the pre-defined cut-off, it is not significant, and it is not associated with the outcome under the study conditions.

2. This is interesting that among 120 subjects, none reported as a smoker while the prevalence of smoking among women reported to range from 10% to 40% (BMC Res Notes. 2015; 8: 469.)

3. One of the outcomes of interest was maternal pregnancy complications in addition to fetal development. What was the rationale for measuring the trace elements at delivery, i.e., end of pregnancy which represents the status of exposure after the presence of the adverse outcome? The subjects in this study are not severely deficient that we can assume a low level at early and mid-pregnancy. Furthermore, iodine deficiency has a high impact during fetal and infant developmental stages.

4. What was the CV for UIC measurements?

5. P3L123-125: If the authors matched cases and controls by age, why did they adjust again for maternal age?

6. In Table 1, the means per group has been presented and the mean difference is not adjusted. Only adjusted p-value has been reported. Table 1 should be characteristics of the study participants, including the mean of measurements. For the adjusted model, the adjusted mean difference obtained from the GLM model with 95% CI should be obtained and reported for pregnancy relative to non-pregnancy. Same for Table 2. For characteristics, it is recommended to report medians and quartiles as there seems to be some non-normal distributions and also the fact that UIC recommendation is based on the median.

7. For thyroid function test, the selected control group (non-pregnant women) is not helpful and clinically meaningful. Thyroid function is different per trimester and also in comparison with pre-pregnancy.

8. Table 3 shows that 48/80 pregnant women (60%) had pregnancy complications. This is a very high rate and does not seem to represent a sample from the general population.

9. One major confounder is weight gain during pregnancy which was not considered in the analyses. At least they could have used pre-delivery BMI or BMI at first pregnancy visit. Only in table 5, at the subtitle for the first time, the authors state the adjustment for BMI which is not clear what time in pregnancy the BMI referred to.

10. Figure 1. The average intake should be provided per study groups. Furthermore, the authors should specifically provide FFQ-estimated daily/weekly dietary intakes for iron and iodine per study groups.

11. What was the supplement intake status among pregnant women? P2L78-80: “Women taking iron or iodine supplements were excluded from the study.” So, no women were using prenatal multivitamin?

12. P6L187-190: This association should be investigated more carefully. The subjects in this study had low consumption of meat and fish. It has been shown that dairy products do not inhibit nonheme-iron absorption (The American Journal of Clinical Nutrition, Volume 80, Issue 2, August 2004, Pages 404–409).

12. Table 5 presents the association of the intake of different foods and beverages on the status of iron and iodine. The need and recommendations for nutrition in non-pregnant women are different from pregnant women. Lumping all subjects for such an association is not appropriate.

Minor issues:

1. P2L56: 47.7% in non-pregnant women? Adults? Pregnant women? The provided link is not working.